# Multiple Cutaneous Manifestations in ANCA-Positive Eosinophilic Granulomatosis with Polyangiitis before and after Biologic Therapy: Clinical and Histopathologic Characterization of a Paradigmatic Case

**DOI:** 10.3390/jcm11247429

**Published:** 2022-12-15

**Authors:** Giorgia Carnicelli, Alvise Sernicola, Vito Gomes, Giulia Cundari, Stefania Trasarti, Roberta Priori, Teresa Grieco

**Affiliations:** 1Department of Radiology, Humanitas Research Hospital IRCCS, Rozzano, 20089 Milan, Italy; 2Dermatology Unit, Department of Clinical Internal Anesthesiological and Cardiovascular Sciences, “Sapienza” University of Rome, 00161 Rome, Italy; 3Department of Anatomy and Pathology, Ospedale San Filippo Neri, 00135 Rome, Italy; 4Department of Radiological Sciences, Oncology and Pathology, “Sapienza” University of Rome, 00161 Rome, Italy; 5Department of Cellular Biotechnology and Hematology, “Sapienza” University of Rome, 00161 Rome, Italy; 6Rheumatology Unit, Department of Applied Clinical and Medical Therapy, “Sapienza” University of Rome, 00161 Rome, Italy

**Keywords:** eosinophilic granulomatosis with polyangiitis, ANCA, cutaneous manifestation, vasculitis, rituximab

## Abstract

Eosinophilic granulomatosis with polyangiitis (EGPA) is a rare immune-mediated vasculitis associated with anti-neutrophil cytoplasmic antibodies (ANCAs). Having systemic and possibly severe involvement, a prompt recognition of its clinical features is crucial to achieve favorable patient outcomes. Although cutaneous manifestations represent key elements, these still remain poorly characterized. We report a case of ANCA-positive EGPA presenting with palpable purpura, livedo reticularis, and pemphigoid-like lesions that was successfully treated with glucocorticoid therapy and rituximab. This report portrays the evolution of cutaneous lesions in ANCA-positive EGPA and demonstrates how dermatologic signs may represent indicators of active disease, allowing for timely diagnosis and for the monitoring of disease activity during treatment.

## 1. Introduction

Eosinophilic granulomatosis with polyangiitis (EGPA), formerly Churg Strauss syndrome, is a rare ANCA-associated vasculitis (AAV). The hallmark of this disease is eosinophilic inflammation, which targets mostly the pulmonary, renal, cutaneous, and nervous systems [1]. Distinct patterns of occurrence are described for MPO-ANCA-positive and seronegative EGPA [2,3,4]. Being a rare disease, the clinical evidence for its understanding is still scarce, especially in regard to skin involvement [5,6].

## 2. Case Description

A 60-year-old male was referred to our dermatologic consultation in September 2020 for the new appearance of widespread cutaneous lesions. The patient had been admitted to the hematology unit of our hospital two months before due to worsening weakness, persistent low-grade fever, and cough. Blood values were suggestive of a possible myeloproliferative syndrome or hematologic malignancy: complete blood count was remarkable for hypereosinophilia (12,170 cells/mL, 34%), anemia (Hgb: 9.7 mg/dL), and leukocytosis (21,660 cells/mL), along with a systemic inflammatory status (ESR 120 mm/h, CRP 78,100 ug/L), polyclonal hypergammaglobulinemia (32%), and total serum IgE count of 672 IU/mL. Urinalysis revealed microscopic proteinuria (50 mg/dL), without any elevation in serum creatinine or in other markers. Personal medical history was positive for a diagnosis of asthma and chronic sinusitis for which he was not taking medications. The patient also reported a recent weight loss of more than 7 kg and episodes of recurrent respiratory infections treated with unspecified systemic antibiotics. 

At the time of first presentation, skin examination showed multiple tense bullous lesions affecting the lower half of the legs. The largest measured 35 mm, had serous content, and was surrounded by normal-appearing skin, resembling a “pemphigoid-like” blistering disorder. Mucosal surfaces were spared. Co-existing lesions were diffuse purpuric-hemorrhagic macules affecting the surface of the lower legs bilaterally, suggestive of a vasculitis-associated cutaneous pattern. Within one week from the first presentation, concomitant with a flare-up of constitutional symptoms, the dermatologic picture evolved into a serpiginous, purple-colored rash affecting the trunk and the upper and lower limbs, which was recognized as generalized livedo reticularis (Figure 1).

## 3. Diagnostic Assessment

Skin biopsies performed in the affected areas revealed epidermal bullous detachment at the basement membrane zone. Multiple foci of vasculitis, characterized by parietal fibrinoid necrosis and eosinophil infiltration, were present in the mid-dermis and hypodermis, consistent with eosinophilic necrotizing polyangiitis with bullous manifestation [7] (Figure 2). A bone marrow biopsy was performed in August 2020, and the FISH analysis of PDGFRα, FIP1L1, and CHIC2 gene rearrangements was unremarkable, excluding the diagnosis of eosinophilic malignancies. Autoantibody panel revealed a strong positivity for anti-myeloperoxidase anti-neutrophil cytoplasmic antibodies (MPO-ANCA, 48 IU/mL), which could be related to EGPA and other antibody-associated vasculitides. MPO-ANCAs are detected in only 30–35% of patients with EGPA (low sensitivity) but have a good specificity, estimated to be around 95% for this disease [8,9].

On the ground of cutaneous presentation, clinical, and histopathological findings, a diagnosis of EGPA was made, in line with the criteria of the American College of Rheumatology and those proposed by Robson et al. [10,11].

The patient was then investigated for internal organ involvement: total body CT showed pulmonary diffuse ground glass opacities and apical pseudo-nodular lesions, and inflammatory thickening and erosion of paranasal sinuses was also found. Cardiac 1.5 T MRI demonstrated ventricular hypokinesis, homogeneous thickening of pericardial leaflets, and diffuse non-ischemic interstitial fibrosis, suggestive of eosinophilic myocarditis in the chronic phase (Figure 3). Electroneurography reported a length-dependent, sensory-motor polyneuropathy in the initial stages. The Birmingham Vasculitis Activity Score (BVAS) at admission was 24.

Considering symptom severity, with evidence of severe active disease and ANCA positivity, the patient was transferred to the Rheumatology Unit in October 2020 where i.v. methylprednisolone 30 mg/day was promptly started, and once-a-week infusion of rituximab at 375 mg/m^2^ was administered for four weeks. Methylprednisolone was de-escalated to 8 mg/day during the following month.

At the 10-week follow-up (January 2021), the patient reported an improvement in his general conditions and a body weight gain of five kilograms; the BVAS score at the time was five, demonstrating a good control over the disease. The blood test was normalized, showing hypogammaglobulinemia (9%) and negativity for p-ANCA. Parallel to the clinical improvement, the dermatological picture underwent remission since the start of rituximab: all the purpuric and the bullous lesions regressed, with the latter leaving atrophic scars. Generalized livedo reticularis evolved into light brown pigmented lesions, indicative of non-active disease. We documented a recognizable sequence of cutaneous lesions developing during disease flares; throughout the regular follow-up checks until the present time, we observed an association between the appearance of new skin lesions and the onset of organ involvement. Finally, follow-up urinalysis was always negative for proteinuria and the patient did not consent to a kidney biopsy to rule out renal involvement. The timeline of interventions and the outcomes for this case are summarized in Table 1.

## 4. Discussion

EGPA is to date an incompletely characterized disease due to the heterogeneous presentation and low incidence. The immunopathology is complex, having features of small vessel vasculitis often overlapping with eosinophilic inflammation. In addition, Th1, Th-2, and Th-17-mediated immunities are described [12]. A landmark genetic study by Lyons et al. demonstrates that seropositive and seronegative EGPA cases correspond to two distinct genetic signatures and can be considered separate disease entities. While MPO-positive EGPA is closely related to other AAVs for its genetic profile, MPO-negative disease has molecular features similar to asthma and may derive from a pre-existing barrier dysfunction and hyper-eosinophilia [2,4]. Recognizing this difference is important for the sake of treatment: while ANCA-positive EGPA is responsive to rituximab (around 80% response rate), ANCA-negative is less sensitive to this drug [2,13,14,15,16]. 

Dermatologic manifestations are estimated to occur in around 50% of patients, representing a relatively frequent type of involvement. Although different clinical pictures have been described in EGPA (purpura and hemorrhagic lesions, cutaneous nodules, and more rarely sero-hematic bullae, livedo reticularis, and urticaria), still no distinction has been made between seropositive and seronegative diseases [12,13,17]. We firstly describe the cutaneous features of a vasculitic disease phenotype in a case of seropositive EGPA, which satisfied both the ACR and the newly proposed Robson diagnostic criteria [10,11,18]. Small vessel vasculitis is mostly associated with hemorrhagic alveolitis, rapidly progressive glomerulonephritis, and mononeuritis multiplex [2,12].

We identified a seropositive-specific evolution of skin lesions, arising with purpura and bullous manifestations and progressing to generalized livedo reticularis. Similarly, Ishibashi et al. report a higher incidence of purpura and hemorrhagic bullae in ANCA-positive EGPA [6].

Current guidelines endorse the use of rituximab for the treatment of severe active disease in patients with ANCA-positive EGPA as an alternative to the traditionally used cyclophosphamide. While the latter would be preferential in cases with active cardiac involvement or with negative ANCA, rituximab is an effective corticosteroid-sparing agent, which is associated with durable remission and is safe in patients with renal involvement [16,19]. In line with this evidence, our patient experienced a significant amelioration of cutaneous lesions upon an induction therapy with rituximab and a dramatic improvement in disease activity (BVAS = 5, remission).

We hypothesize that cutaneous manifestations may accurately reflect disease burden and internal organ involvement: we observed a direct correlation between cutaneous disease activity and BVAS score. Hemorrhagic lesions are already described as indicators of flare-ups and are associated with more severe involvement [20].

It is our opinion that dermatologic manifestations should always be monitored in EGPA patients, constituting important red flags for active disease. Our report provides a comprehensive dermatologic characterization of ANCA-positive EGPA, ultimately supporting the application of rituximab in this disease phenotype.

## 5. Take-Away Lessons from the Case

Eosinophilic granulomatosis with polyangiitis (EGPA) is a rare ANCA-associated vasculitis with systemic involvement: dermatologic manifestations are heterogeneous, with no distinction available between seropositive and seronegative diseases.We describe a specific pattern of cutaneous manifestations observed in ANCA-positive EGPA, characterized by a sequence of hemorrhagic and bullous lesions, followed by livedo reticularis.We observed that the activity of cutaneous disease correlates directly with BVAS and reflects internal organ involvement.Cutaneous lesions should always be monitored, as possible early indicators of flares of disease.

## Figures and Tables

**Figure 1 jcm-11-07429-f001:**
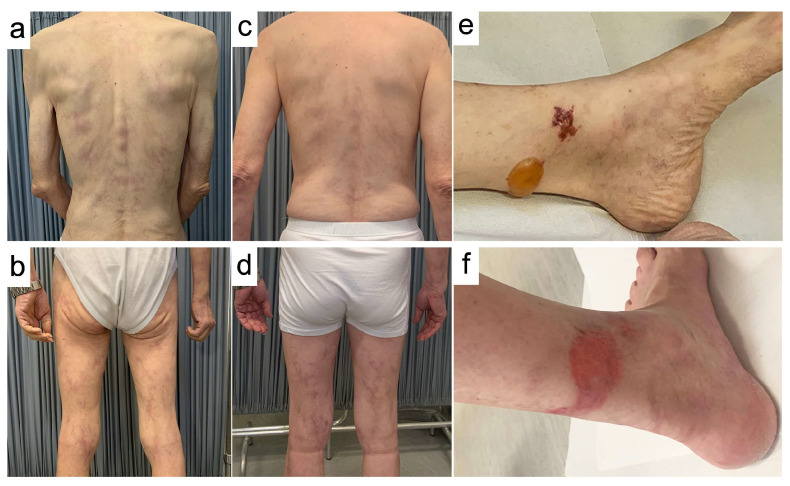
Clinical examination at the time of presentation (**a**,**b**,**e**), and after one month of therapy with rituximab (**c**,**d**,**f**). Generalized livedo reticularis, with serpiginous red-purple lesions involving the whole surface of the body (**a**,**b**). Purpuric-hemorrhagic lesions and “pemphigoid-like” serous bulla (d = 3.5 cm) involving the lower third of the leg (**e**). Improvement of generalized livedo reticularis with evolution into brown pigmentary lesions, along with sensible weight gain (**c**,**d**) and remission of the lesions in the lower leg (**f**).

**Figure 2 jcm-11-07429-f002:**
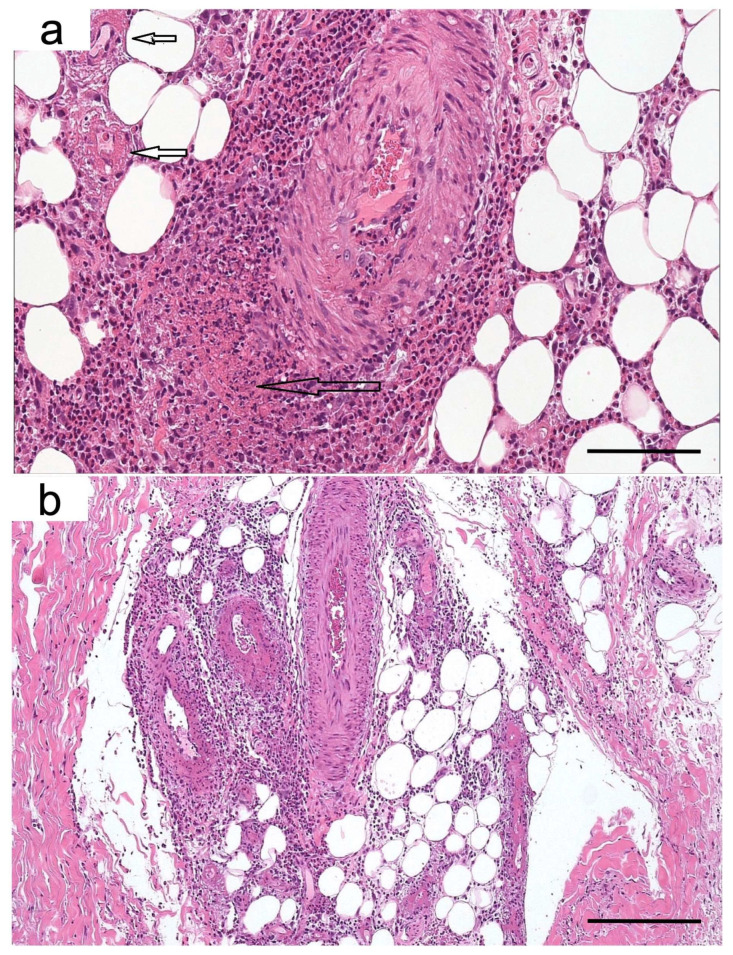
Skin biopsy. Foci of vasculitis characterized by parietal fibrinoid necrosis (small arrows) and a small/medium caliber arteriole with subintimal infiltration of neutrophils and abundant eosinophils (large arrow) (Hematoxylin and eosin, scale bar 100 µm) (**a**). Arteritis and vasculitis in vessels of different caliber (Hematoxylin and eosin, scale bar 250 µm) (**b**).

**Figure 3 jcm-11-07429-f003:**
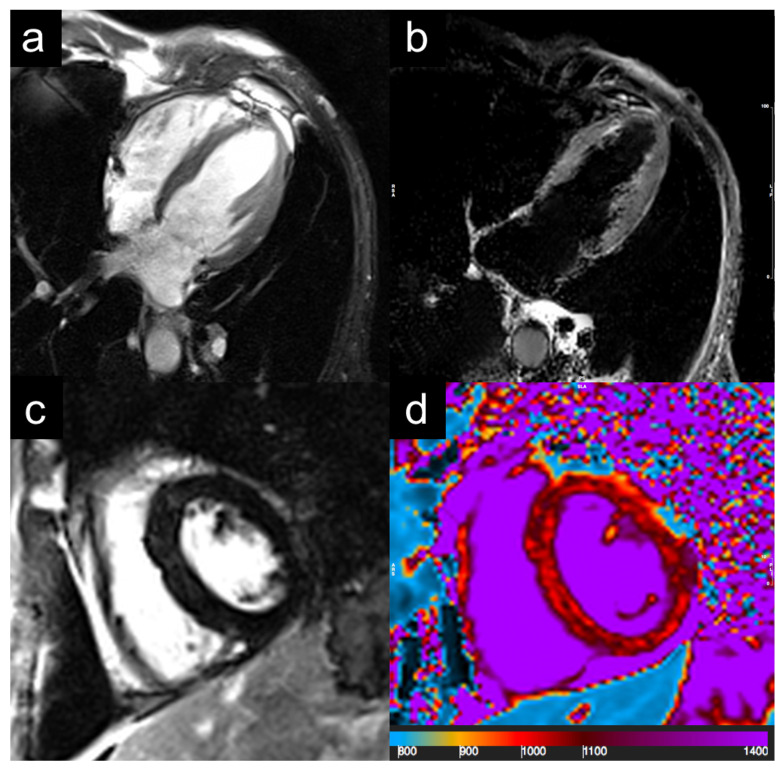
Cardiac magnetic resonance imaging. Cine-images are evidenced of hypokinesia of apical segments (**a**). Absence of myocardial edema and focal myocardial fibrosis at STIR or LGE imaging (**b**,**c**). Native T1 mapping show pericardial effusion and diffuse increase in T1 mapping values (1050–1150 ms; normal values: 950–1030 ms), suggesting diffuse interstitial fibrosis (**d**). STIR: short tau inversion recovery; LGE: late gadolinium enhancement.

**Table 1 jcm-11-07429-t001:** Timeline of interventions and outcomes.

Timeline	Medical History and Past Interventions
2019	Diagnosis of asthma and chronic sinusitis Antibiotic treatment for recurrent respiratory infections
	**Diagnostic testing and interventions**
July 2020	Onset of symptoms - Hospitalization due to weakness, persistent low-grade fever, and cough - Lab tests: eosinophils 12,170 cells/mL, 34%; Hgb: 9.7 mg/dL; leukocytes 21,660 cells/mL; ESR 120 mm/h; CRP 78,100 µg/L; polyclonal hypergammaglobulinemia 32%; total serum IgE 672 IU/mL; and proteinuria 50 mg/dL - BMB (August 2020): FISH for PDGFRα, FIP1L1, CHIC2 rearrangements negative
September 2020	Diagnosis of EGPA - Dermatology consultation for widespread cutaneous lesions - Skin biopsy: eosinophilic necrotizing vasculitis - Lab tests: MPO-ANCA 48 IU/mL
September 2020	Baseline assessment - BVAS 24 - CT scan: pulmonary diffuse ground glass opacities and apical pseudo-nodular lesions; inflammatory thickening and erosion of paranasal sinuses - Cardiac MRI: eosinophilic myocarditis in the chronic phase - ENG: initial sensory-motor polyneuropathy
October 2020	Start of treatment - Methylprednisolone 30 mg/day IV - Rituximab 375 mg/m^2^ IV once-a-week for four weeks - Methylprednisolone de-escalation to 8 mg/day (November 2020)
January 2021	Remission (10-week follow-up) - BVAS 5 - Lab tests: hypogammaglobulinemia 9%, ANCA negative, proteinuria negative
	Maintenance - Rituximab 500 mg IV every 6 months - Methylprednisolone 2 mg/day

Abbreviations: BMB, bone marrow biopsy; BVAS, Birmingham Vasculitis Activity Score; CHIC2, cysteine rich hydrophobic domain 2; CRP, C-reactive protein; CT, computerized tomography; EGPA, eosinophilic granulomatosis with polyangiitis; ENG, electroneurography; ESR, erythrocyte sedimentation rate; FIP1L1, factor interacting with PAPOLA and CPSF1; FISH, fluorescence in situ hybridization; Hgb, hemoglobin; IgE, immunoglobulin E; IV, intravenously; MPO-ANCA, anti-myeloperoxidase anti-neutrophil cytoplasmic antibodies; MRI, magnetic resonance imaging; PDGFRα; platelet-derived growth factor receptor.

## Data Availability

The data presented in this study are available from the corresponding author upon request. The data are not publicly available due to privacy restrictions.

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
