# Peer review of "Multiple Cutaneous Manifestations in ANCA-Positive Eosinophilic Granulomatosis with Polyangiitis before and after Biologic Therapy: Clinical and Histopathologic Characterization of a Paradigmatic Case"

_jcm, 2022, doi:10.3390/jcm11247429_

Round 1

Reviewer 1 Report

The manuscript entitled “Multiple cutaneous manifestations in ANCA-positive eosinophilic granulomatosis with polyangiitis before and after biologic therapy: clinical and histopathologic characterization of a paradigmatic case” by Carnicelli et al. thoroughly describes a case of ANCA-positive EGPA treated with rituximab. The authors propose the monitoring of dermatologic signs for assessing disease activity and response to treatment.

The manuscript is well-written, and overall describes well the patient case, hitting on the most important points. No major comments to address in my opinion, only a few minor points.

Minor comments:

-          In line 140, the authors very rightly suggest that ANCA-negative patients may be more less responsive to rituximab and give two citations. It would be better to add a few more citations that provide evidence of this, as both citations provided have very limited scope and don’t really address this point. If no other citations can be found, I suggest that the sentence be toned down (for instance: “Recognising this could be important for the …)

-          Line 46 I assume this is 21660cells/mL

-          I expect the rituximab treatment was 375mg/m2 not g/m2

-          The authors use the word “affection” on a few occasions. I believe the authors should consider “affliction” or perhaps another word. “Affection” is used to describe something very different.

Author Response

Dear Reviewer, 
thank you for your very appropriate clarifications. We have carefully considered your comments and edited our manuscript accordingly. Our responses are provided below: 
-    Thank you for your constructive comment. We suggest that rituximab could be the option of choice in the light of evidence provided in the new 2021 ACR guidelines on AAVs, and in systematic reviews, which we added to lines 140-142 as citations. Recent evidence supports rituximab to be more effective and to carry a shorter time-to-remission compared to cyclophosphamide in the treatment of ANCA-positive EGPA; notably, it also has a better toxicity profile. Otherwise, cyclophosphamide still demonstrates optimal efficacy in the management of severe seronegative EGPA, and when cardiac involvement is present. For additional clarification, we provide the following statements from the 2021 guidelines: “Ungraded position statement: For patients with active, severe EGPA, either cyclophosphamide or rituximab may be prescribed for remission induction. … Cyclophosphamide can also be considered for patients who are ANCA-negative and have severe neurologic or gastrointestinal manifestations. Rituximab may be considered for patients with positive ANCA results, active glomerulonephritis, prior cyclophosphamide treatment, or those at risk of gonadal toxicity from cyclophosphamide”
Please see lines 140-142 and references 14,15 and 16: “Recognizing this difference is important for the sake of treatment: while ANCA-positive EGPA is responsive to rituximab (around 80% response rate), ANCA-negative is less sensitive to this drug [2,13-16].”
2.    The European Vasculitis Genetics Consortium; Lyons, P.A.; Peters, J.E.; Alberici, F.; Liley, J.; Coulson, R.M.R.; Astle, W.; Bald-ini, C.; Bonatti, F.; Cid, M.C.; et al. Genome-Wide Association Study of Eosinophilic Granulomatosis with Polyangiitis Re-veals Genomic Loci Stratified by ANCA Status. Nat Commun 2019, 10, 5120, doi:10.1038/s41467-019-12515-9.
13.    Wu, E.Y.; Hernandez, M.L.; Jennette, J.C.; Falk, R.J. Eosinophilic Granulomatosis with Polyangiitis: Clinical Pathology Conference and Review. The Journal of Allergy and Clinical Immunology: In Practice 2018, 6, 1496–1504, doi:10.1016/j.jaip.2018.07.001.
14.    Akiyama, M.; Kaneko, Y.; Takeuchi, T. Rituximab for the treatment of eosinophilic granulomatosis with polyangiitis: A systematic literature review. Autoimmunity reviews 2021, 20, 102737. doi:10.1016/j.autrev.2020.102737.
15.    Teixeira, V.; Mohammad, A.J.; Jones, R.B.; Smith, R.; Jayne, D. Efficacy and safety of rituximab in the treatment of eosinophilic granulomatosis with polyangiitis. RMD open 2019, 5, e000905. doi:10.1136/rmdopen-2019-000905.
16.    Chung, S.A.; Langford, C.A.; Maz, M.; Abril, A.; Gorelik, M.; Guyatt, G.; Archer, A.M.; Conn, D.L.; Full, K.A.; Grayson, P.C.; et al. 2021 American College of Rheumatology/Vasculitis Foundation Guideline for the Management of Antineutrophil Cyto-plasmic Antibody–Associated Vasculitis. Arthritis Rheumatol 2021, 73, 1366–1383, doi:10.1002/art.41773.
-     Thank you for your comment, we have changed “21660 cells/m” with “21660 cells/mL” in line 46 (and in Table 1).
-    Thank you, we have changed to “rituximab 375 mg/m2” instead of “rituximab 375 g/m2” in line 106 (and in Table 1).
-    Thank you for the constructive comment: we have replaced the word “affection” with “involvement” in line 22 and 174, and with “occurrence” in line 36.

Reviewer 2 Report

A good written article. Bullous presentation is interesting and treatment sounds.

Author Response

Dear Reviewer,
Thank you for your careful review of our paper and for the encouraging comments.

Reviewer 3 Report

The case on cutaneous manifestations in ANCA-positive eosinophilic granulomatosis with polyangiitis is very well prepared and written in clear language. Cutaneous manifestations are presented with high quality pictures and histopathological findings. The article adds new clinical knowledge in the relationship between the activity of skin lesions and the severity of internal organ involvement.

I do not have special comments.

Author Response

Dear Reviewer, 
Thank you for your efforts on our paper and for your constructive comments.